# Relationships and Qualitative Evaluation between Diffusion-Weighted Imaging and Pathologic Findings of Resected Lung Cancers

**DOI:** 10.3390/cancers12051194

**Published:** 2020-05-08

**Authors:** Katsuo Usuda, Shun Iwai, Aika Yamagata, Atsushi Sekimura, Nozomu Motono, Munetaka Matoba, Mariko Doai, Sohsuke Yamada, Yoshimichi Ueda, Keiya Hirata, Hidetaka Uramoto

**Affiliations:** 1Department of Thoracic Surgery, Kanazawa Medical University, Ishikawa 920-0293, Japan; mhg1214@kanazawa-med.ac.jp (S.I.); aicarby@kanazawa-med.ac.jp (A.Y.); a24seki@kanazawa-med.ac.jp (A.S.); motono@kanazawa-med.ac.jp (N.M.); hidetaka@kanazawa-med.ac.jp (H.U.); 2Department of Radiology, Kanazawa Medical University, Ishikawa 920-0293, Japan; m-matoba@kanazawa-med.ac.jp (M.M.); doaimari@kanazawa-med.ac.jp (M.D.); 3Department of Pathology and Laboratory Medicine, Kanazawa Medical University, Ishikawa 920-0293, Japan; sohsuke@kanazawa-med.ac.jp; 4Department of Pathophysiological and Experimental Pathology, Kanazawa Medical University, Ishikawa 920-0293, Japan; z-ueda@kanazawa-med.ac.jp; 5MRI Center, Kanazawa Medical University Hospital, Ishikawa 920-0293, Japan; keiya@kanazawa-med.ac.jp

**Keywords:** diffusion-weighted magnetic resonance imaging (DWI), magnetic resonance imaging (MRI), lung cancer, pathology, apparent diffusion coefficient (ADC)

## Abstract

For detecting malignant tumors, diffusion-weighted magnetic resonance imaging (DWI) as well as fluoro-2-deoxy-glucose positron emission tomography/computed tomography (FDG-PET/CT) are available. It is not definitive how DWI correlates the pathological findings of lung cancer. The aim of this study is to evaluate the relationships between DWI findings and pathologic findings. In this study, 226 patients with resected lung cancers were enrolled. DWI was performed on each patient before surgery. There were 167 patients with adenocarcinoma, 44 patients with squamous cell carcinoma, and 15 patients with other cell types. Relationships between the apparent diffusion coefficient (ADC) of DWI and the pathology were analyzed. When the optimal cutoff value (OCV) of ADC for diagnosing malignancy was 1.70 × 10^−3^ mm^2^/s, the sensitivity of DWI was 92.0% (208/226). The sensitivity was 33.3% (3/9) in mucinous adenocarcinoma. The ADC value (1.31 ± 0.32 × 10^−3^ mm^2^/s) of adenocarcinoma was significantly higher than that (1.17 ± 0.29 × 10^−3^ mm^2^/s) of squamous cell carcinoma (*p* = 0.012), or (0.93 ± 0.14 × 10^−3^ mm^2^/s) of small cell carcinoma (*p* = 0.0095). The ADC value (1.91 ± 0.36 × 10^−3^ mm^2^/s) of mucinous adenocarcinoma was significantly higher than that (1.25 ± 0.25 × 10^−3^ mm^2^/s) of adenocarcinoma with mucin and that (1.24 ± 0.30 × 10^−3^ mm^2^/s) of other cell types. The ADC (1.11 ± 0.26 × 10^−3^ mm^2^/s) of lung cancer with necrosis was significantly lower than that (1.32 ± 0.33 × 10^−3^ mm^2^/s) of lung cancer without necrosis. The ADC of mucinous adenocarcinoma was significantly higher than those of adenocarcinoma of other cell types. The ADC of lung cancer was likely to decrease according to cell differentiation decreasing. The sensitivity of DWI for lung cancer was 92% and this result shows that DWI is valuable for the evaluation of lung cancer. Lung cancer could be evaluated qualitatively using DWI.

## 1. Introduction

Lung cancer is one of the leading causes of cancer-related deaths and has many patterns of progression and treatment responses. As the imaging method of choice in tumor staging, fluoro-2-deoxy-glucose positron emission tomography/computed tomography (FDG-PET/CT) has been widely adopted. Its maximum standardized uptake value (SUVmax) presents glucose uptake and indicates how aggressive the cancer is. FDG-PET/CT is useful for differentiating malignant from benign pulmonary nodules [1]. However, FDG-PET/CT is likely to yield false-negative results for small volumes of metabolically active tumors [2] or well-differentiated pulmonary adenocarcinoma [3], and false-positive results for inflammatory nodules [4].

For the last two decades, magnetic resonance imaging (MRI) of the staging of lung cancer has been narrowly available in the cases of chest wall invasion or mediastinum invasion of lung cancer partly due to the report of Webb et al. [5] of the Radiologic Diagnostic Oncology Group published in 1991. The technology of MRI has developed dramatically. Diffusion-weighted magnetic resonance imaging (DWI) has been used for detecting the restricted diffusion of water molecules. The principle of DWI is the random motion of water molecules in biological tissues [6]. Its apparent diffusion coefficient (ADC) value presents a quantitative parameter of the diffusion of water molecules in biological tissues, and the ADC of malignant tumors is significantly lower than that of normal tissues or benign lesions [7]. The magnetic resonance (MR) signal intensity of pulmonary cancer is significantly higher than that of benign lesions [8]. A meta-analysis indicated that DWI could be used to differentiate malignant from benign pulmonary lesions [9]. Two articles of meta-analysis have shown that DWI was useful for the evaluation of the N factor of lung cancer [10,11]. For nodal assessment in non-small cell lung cancer, Peerlings et al. [10] showed the high diagnostic capability of DWI (sensitivity 0.87, specificity 0.88). DWI can differentiate benign from malignant lesions in the lung [9,12], in the thorax [13], in the prostate [14], in the breast [15], and in the liver [16].

DWI possesses great potential for monitoring treatment response in cancer patients shortly after the initiation of radiotherapy [17]. Functional evaluation of DWI was more useful than that of CT for the response evaluation of chemotherapy and/or radiotherapy to recurrent tumors of the lung [18]. MR functional imaging offers valuable information about tumor tissue, tissue architecture, cellular biomarkers related to the hepatocellular functions, and tissue vascularization profiles related to tumor and tissue biology [19]. Maximum whole tumor ADC values may be available for differentiating luminal from other molecular subtypes of breast cancer [20].

In this article, we analyze relationships between DWI and the pathological findings of resected lung cancers and got information about DWI for the pathologic characteristics of lung cancer. 

## 2. Results

Chest CT, FDG-PET/CT, DWI, ADC map, and pathologic hematoxylin and eosin stain are presented according to lepidic adenocarcinoma (Figure 1), mucinous adenocarcinoma (Figure 2), papillary adenocarcinoma (Figure 3) and squamous cell carcinoma (Figure 4).

When the optimal cutoff value (OCV) of ADC for diagnosing malignancy was 1.70 × 10^−3^ mm^2^/s [21], the sensitivity of DWI was 92.0% (208/226) (Table 1). For pathologic cell types, the sensitivity of DWI was 91.6% (153/167) in adenocarcinoma, 95.4% (42/44) in squamous cell carcinoma, 66.6% (2/3) in large cell neuroendocrine carcinoma (LCNEC), 66.6% (2/3) in large cell carcinoma, and 100% (6/6) in small cell carcinoma. For subtypes of adenocarcinoma, the sensitivity of DWI for mucinous adenocarcinoma was 33.3% (3/9), which was significantly lower than 94.4% (51/54) for acinar adenocarcinoma, 94.6% (53/56) for papillary adenocarcinoma, 92.8% (26/28) for lepidic adenocarcinoma, 100% (7/7) for micropapillary adenocarcinoma, or 100% (12/12) of solid adenocarcinoma. 

The ADC value by pathologic cell type of lung cancer is presented in Figure 5. The ADC value (1.31 ± 0.32 × 10^−3^ mm^2^/s) of adenocarcinoma was significantly higher than that (1.17 ± 0.29 × 10^−3^ mm^2^/s) of squamous cell carcinoma (*p* = 0.012), or (0.93 ± 0.14 × 10^−3^ mm^2^/s) of small cell carcinoma (*p* = 0.0095). The ADC value (1.62 ± 0.40 × 10^−3^ mm^2^/s) of LCNEC was significantly higher than that (1.17 ± 0.29 × 10^−3^ mm^2^/s) of squamous cell carcinoma (*p* = 0.016), or (0.93 ± 0.14 × 10^−3^ mm^2^/s) of small cell carcinoma (*p* = 0.011).

The ADC value by pathologic subtype of adenocarcinoma is shown in Figure 6. The ADC value (1.91 ± 0.36 × 10^−3^ mm^2^/s) of mucinous adenocarcinoma was significantly higher than that (1.31 ± 0.28 × 10^−3^ mm^2^/s) of acinar adenocarcinoma (*p* < 0.0001), (1.29 ± 0.30 × 10^−3^ mm^2^/s) of papillary adenocarcinoma (*p* < 0.0001), (1.28 ± 0.28 × 10^−3^ mm^2^/s) of lepidic adenocarcinoma (*p* < 0.0001), (1.14 ± 0.20 × 10^−3^ mm^2^/s) of micropapillary adenocarcinoma (*p* = 0.0002), or (1.12 ± 0.20 × 10^−3^ mm^2^/s) of solid adenocarcinoma (*p* < 0.0001). 

The ADC value (1.91 ± 0.36 × 10^−3^ mm^2^/s) of mucinous adenocarcinoma was significantly higher than that (1.25 ± 0.25 × 10^−3^ mm^2^/s) of adenocarcinoma with mucin, or (1.24 ± 0.30 × 10^−3^ mm^2^/s) of other cell types (Figure 7). The ADC (1.11 ± 0.26 × 10^−3^ mm^2^/s) of lung cancer with necrosis was significantly lower than that (1.32 ± 0.33 × 10^−3^ mm^2^/s) of lung cancer without necrosis (*p* = 0.0001) (Figure 8). For cell differentiation, the ADC value was 1.36 ± 0.35 × 10^−3^ mm^2^/s in well differentiation (G1), 1.24 ± 0.28 × 10^−3^ mm^2^/s in moderate differentiation (G2), 1.12 ± 0.19 × 10^−3^ mm^2^/s in poor differentiation (G3), and 1.18 ± 0.43 × 10^−3^ mm^2^/s in undifferentiation (G4). There is a correlation between the decrease in ADC value and the decrease in cell differentiation. 

## 3. Discussion

A concise update of lung cancer staging, supported by the European Society of Thoracic Surgeons and the American College of Chest Physicians, expressed that DWI can differentiate benign from malignant lymph nodes and showed that diffusion MRI has equal sensitivity to PET-CT (0.75 versus 0.72, respectively) but higher specificity (PET-CT 0.89 versus MRI 0.95) [22]. In our experience of hilar and mediastinal lymph nodes in lung cancer [23], DWI correctly diagnosed N staging in 144 carcinomas (90%) but incorrectly diagnosed N staging in 16 (10%) (3 (1.9%) had overstaging, 13 (8.1%) had understaging). PET-CT correctly diagnosed N staging in 133 carcinomas (83.1%) but incorrectly diagnosed N staging in 27 (16.8%) (4 (2.5%) had overstaging, 23 (14.4%) had understaging). The maximum diameter of metastatic lesions in lymph nodes was 3.0 ± 0.9 mm in 21 lymph node stations not detected by either DWI or PET-CT, 7.2 ± 4.1 mm in 39 detected by DWI, and 11.9 ± 4.1 mm in 24 detected by PET-CT. Therefore, DWI could detect significantly smaller lymph node metastases than PET-CT could. DWI was reported to be superior to FDG-PET in the detection of primary lesions and the nodal assessment of non-small cell lung cancers [21].

In the Japanese lung cancer practice guidelines published in 2018 [24], MRI was recommended partly for the diagnosis of lung cancer. Recommendation of the usage of DWI was discussed in 2008 during the International Society for Magnetic Resonance in Medicine meeting held in Toronto [25]. Concerns about the lack of understanding of DWI were summarized in the meeting report [25]. 

The optimal cutoff value is very useful for distinguishing malignancy from benignity but would change based on the patients of a study. There are two articles which compare the diagnostic capability of DWI with that of FDG-PET/CT for pulmonary nodules and masses [12,26]: in one, the sensitivity and the accuracy of DWI were significantly higher [12]; in the other, the sensitivity of DWI was significantly higher [26] than that of FDG-PET/CT. Based on our assessment of DWI for pulmonary nodules and masses (143 lung cancers, 17 metastatic lung tumors, and 29 benign pulmonary nodules and masses) [26], the sensitivity (80.0%) of DWI was significantly higher than that (70.0%) of FDG-PET/CT. The specificity (65.5%) of DWI was equal to that (65.5%) of FDG-PET/CT. The accuracy (77.8%) of DWI was not significantly higher than that (69.3%) of FDG-PET/CT for pulmonary nodules and masses. The results imply DWI has higher potential than FDG-PET/CT in assessing pulmonary nodules and masses. There were three meta-analyses of DWI for the differential diagnosis of malignancy and benignity for pulmonary nodules and masses [9,27,28]. All of the meta-analyses concluded that DWI could differentiate malignancy from benignity for pulmonary nodules and masses. However, DWI is used limitedly restrictive in the lungs and is not popular yet. MRI eradicates not only contrast mediums but also radiation exposure, and it is suitable and ideal for the examination of children and pregnant women. In the future, we believe MRI will be more available for lung cancer assessment because CT and FDG-PET/CT have some risk of radiation exposure which must be explained and is not expected by patients.

The ADC value of adenocarcinoma was significantly higher than that of either squamous cell carcinoma or small cell carcinoma, which shows that the tissue cellularity of squamous cell carcinoma or small cell carcinoma would be higher than that of adenocarcinoma. Through DWI examination we realized that we could detect histopathological necrosis and mucinous areas in lung cancer and the ADC values could be related to the pathological structure. 

One of the pulmonary lesions which presented restricted diffusion and lower ADC values in DWI was a pulmonary abscess with pathological necrosis. The heavily impeded mobility of pus may have been caused by its high cellularity and viscosity and shows the low ADC values [29]. Abscesses and thrombi impede the diffusivity of water molecules because they have a hyperviscous nature [30,31]. The median ADC value (0.877 × 10^−3^ mm^2^/s) of abscesses was significantly lower than that (2.118 × 10^−3^ mm^2^/s) of phlegmon (*p* < 0.001) and that (3.008 × 10^−3^ mm^2^/s) of edema (*p* < 0.01) [32]. In our article, the ADCs of lung cancers were distributed over 0.9 to 1.6 × 10^−3^ mm^2^/s. As a result, the ADC of a lung cancer with necrosis became lower than that of a lung cancer without necrosis because the ADC value of the abscess was lower. If the ADC value of another tumor was lower than that of the abscess, the ADC of a tumor with necrosis would become higher than that of a tumor without necrosis. In DWI, 22% of benign lesions exhibited restricted diffusion in images with high b-values [33]. The articles for the characteristics of abscesses and thrombi can show false-positive results in DWI for some benign pulmonary nodules and masses with abscesses. 

On the other hand, mucinous carcinomas were usually hypointense and showed higher ADC values, which could be misdiagnosed as benign lesions in DWI [26]. Mucinous carcinomas had lower DWI signal intensities and higher ADC values than tubular adenocarcinomas in the ano-rectal region, because mucinous carcinomas present lower cellularity than tubular adenocarcinomas [34].

DWI and FDG-PET/CT have their own advantages [35]. DWI provides quantitative information regarding tissue cellularity and the diffusion of water molecules which are not necessarily related to cancer aggressiveness. FDG-PET/CT expresses glucose uptake and presents the aggressiveness of neoplasia of inflammation. Gallivanone et al. [36] reported that FDG-PET/CT predicted patient prognosis and DWI response to neoadjuvant chemotherapy, and both examinations provide useful complementary information for biological characterization and neoadjuvant chemotherapy response prediction in breast cancer. DWI could be added for the differential diagnosis of benign lesions and malignant lesions of not only lungs but also other organs. The differential diagnosis could become possible if we understand the strengths and weaknesses of DWI. 

We should keep in mind that the study had three limitations. First, it was a retrospective study and was conducted at a single institution. Second, our ADC measurements were repeated three times and the minimum ADC value was obtained. There is no consensus for the optimal DWI techniques and image analysis procedure in the literature, including region of interest (ROI) size and placement. Third, in some cases, image quality could make calculating the true ADC value more difficult. This might be a limitation of this technique compared to PET. 

Further studies would be necessary to evaluate the performance of DWI for pathological characteristics.

## 4. Patients and Methods

### 4.1. Eligibility

The study protocol for examining DWI and FDG-PET/CT in patients with lung cancer was approved by the ethical committee of Kanazawa Medical University (the approval number: No. I302). Written informed consent for MRI and a pathological examination of resected materials were obtained from each patient after discussing the risks and benefits of the examinations with their surgeons. 

### 4.2. Patients

There were three patients who were excluded due to lower imaging quality. Finally, 226 patients with primary lung cancer were enrolled in this study (Table 2). They underwent DWI and FDG-PET/CT examination before pulmonary resection with nodal dissection from May 2009 to February 2014. Our previous article [26] dealt with 189 patients with pulmonary nodules and masses prospectively that underwent FDG-PET/CT and DWI. There were 143 lung cancers, 17 metastatic lung tumors, and 29 benign pulmonary nodules and masses. The 143 lung cancers were included in this study. None of the patients had received prior treatment. Out of the 226 patients, 133 were male and 93 were female. Their mean age was 68 years old (range 37 to 85).

There were 167 adenocarcinomas, 44 squamous cell carcinomas, 5 small cell carcinomas, 3 large cell neuroendocrine carcinoma (LCNEC), 3 large cell carcinomas, and 4 carcinomas of other cell types. Three LCNECs were all combined LCNEC and adenocarcinoma. TNM classification and the lymph node stations of lung cancer were classified according to the new definition of UICC (Union for International Cancer Control) 7 [37]. There were 77 pathological T1a (pT1a) carcinomas, 41 pT1b carcinomas, 66 pT2a carcinomas, 13 pT2b carcinomas, 26 pT3 carcinomas, and 3 pT4 carcinomas. There were 179 pathological pN0 (pN0) carcinomas, 30 pN1 carcinomas, and 17 pN2 carcinomas. There were 112 pathological Stage IA (pStage IA), 49 pStage IB, 20 pStageIIA, 13 pStage IIB, 25 pStage IIIA, 1 pStage IIIB, and 6 pStage IV.

### 4.3. MR Imaging

All MR images were produced with a 1.5 T superconducting magnetic scanner (Magnetom Avanto; Siemens, Erlangen, Germany) with two anterior six-channel body phased-array coils and two posterior spinal clusters (six-channels each). The conventional MR images consisted of a coronal T1-weighted spin-echo sequence and coronal and axial T2-weighted fast spin-echo sequences. DWIs using a single-shot echo-planar method were applied with a slice thickness of 6mm under SPAIR (spectral attenuated inversion recovery) with a respiratory triggered scan with the following parameter: TR/TE/flip angle, 3000–4500/65/90; diffusion gradient encoding in three orthogonal directions; *b* value = 0 and 800 s/mm^2^; field of view, 350 mm; matrix size, 128 × 128. After image reconstruction, a two-dimensional (2D) round or elliptical region of interest (ROI) was drawn on the lesion which was detected visually on the ADC map with reference to T2-weighted or CT image by a radiologist (M.D.) with 25 years of MRI experience who was unaware of the patients’ clinical data. The procedures were repeated three times and the minimum ADC value was obtained. The radiologist (M.D.) and one pulmonologist (K.U.) with 28 years of experience evaluated the MRI data. They eventually reached the same consensus. The OCV of ADC for diagnosing malignancy in DWI was determined to be 1.70 × 10^−3^ mm^2^/s using the receiver operating characteristics curve as previously reported [21]. 

### 4.4. Pathological Findings

The parameters of pathological findings were adopted according to pathological reports: cell type, sub-cell type, T factor, N factor, pathological stage, cell differentiation, presence of mucin, and presence of necrosis. 

### 4.5. Statistical Analysis

The data is expressed as the mean ± standard deviation. A two-tailed Student-*t* test was applied for comparison of ADC values in several prognostic factors. The statistical analyses were performed using the computer software program StatView for Windows (Version 5.0; SAS Institute Inc. Cary, NC, USA). A *p* value of <0.05 was considered statistically significant. 

## 5. Conclusions

The sensitivity of DWI for lung cancer was 92% and DWI would be a useful tool for clinical diagnosis and evaluation of lung cancer. DWI would have an advantage in qualitative evaluation of lung cancer.

## Figures and Tables

**Figure 1 cancers-12-01194-f001:**
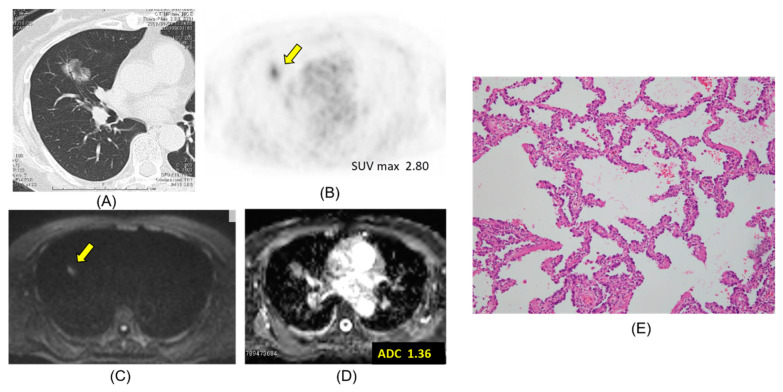
Lepidic adenocarcinoma. The ADC of the carcinoma was 1.36 × 10^−3^ mm^2^/s. (**A**) CT, (**B**) PET-CT, (**C**) DWI, (**D**) ADC map, (**E**) Pathology (Hematoxylin and Eosin Staining) ×100.

**Figure 2 cancers-12-01194-f002:**
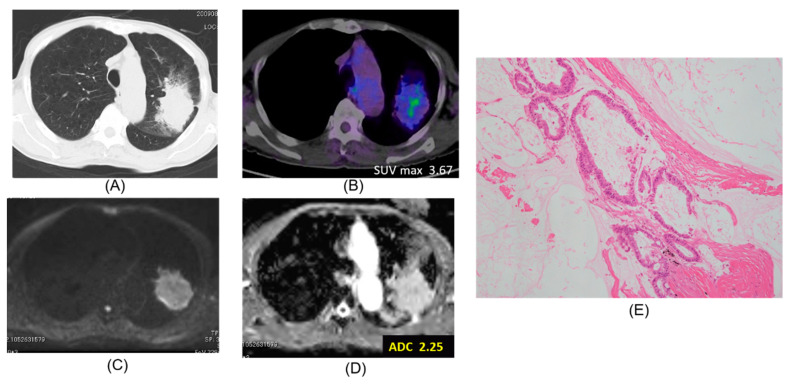
Mucinous adenocarcinoma. The ADC of the carcinoma was 2.25 × 10^−3^ mm^2^/s. (**A**) CT, (**B**) PET-CT, (**C**) DWI, (**D**) ADC map, (**E**) Pathology (Hematoxylin and Eosin Staining) ×100.

**Figure 3 cancers-12-01194-f003:**
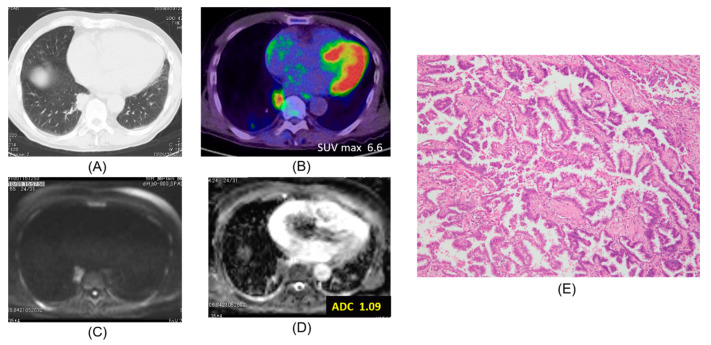
Papillary adenocarcinoma. The ADC of the carcinoma was 1.09 × 10^−3^ mm^2^/s. (**A**) CT, (**B**) PET-CT, (**C**) DWI, (**D**) ADC map, (**E**) Pathology (Hematoxylin and Eosin Staining) ×100.

**Figure 4 cancers-12-01194-f004:**
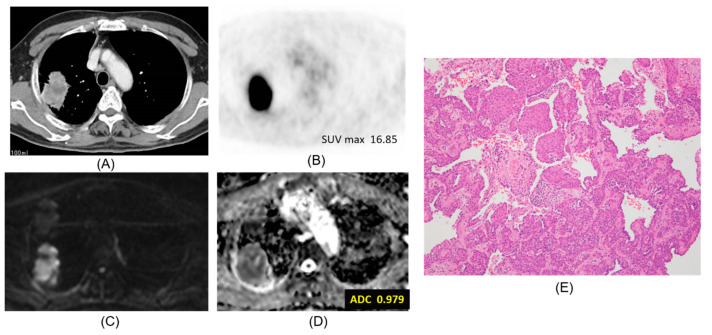
Squamous cell carcinoma. The ADC of the carcinoma was 0.979 × 10^−3^ mm^2^/s. (**A**) CT, (**B**) PET-CT, (**C**) DWI, (**D**) ADC map, (**E**) Pathology (Hematoxylin and Eosin Staining) ×100.

**Figure 5 cancers-12-01194-f005:**
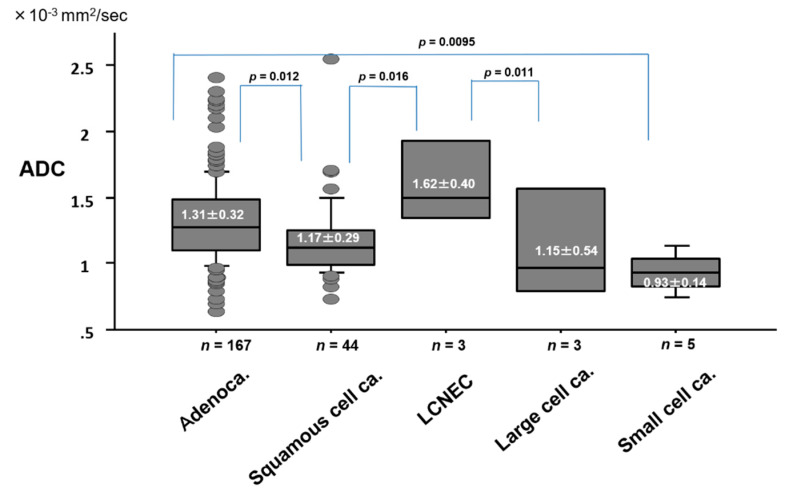
ADC value by pathologic cell type of lung cancer.

**Figure 6 cancers-12-01194-f006:**
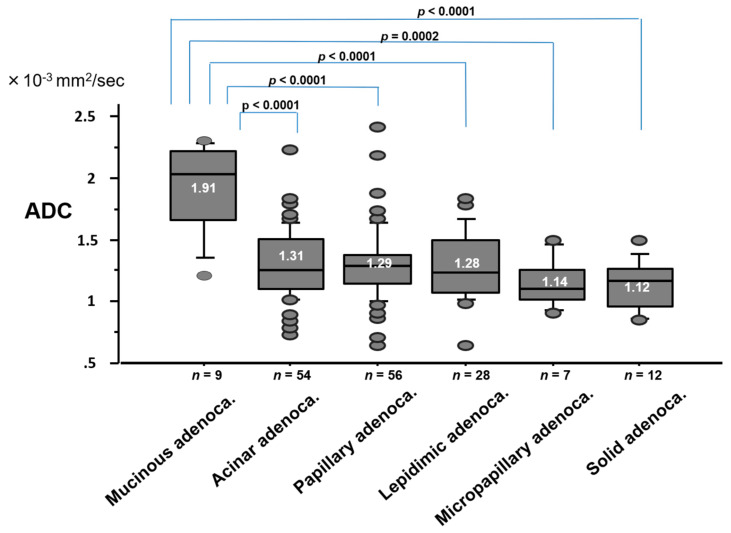
ADC value by pathologic subtype of adenocarcinoma.

**Figure 7 cancers-12-01194-f007:**
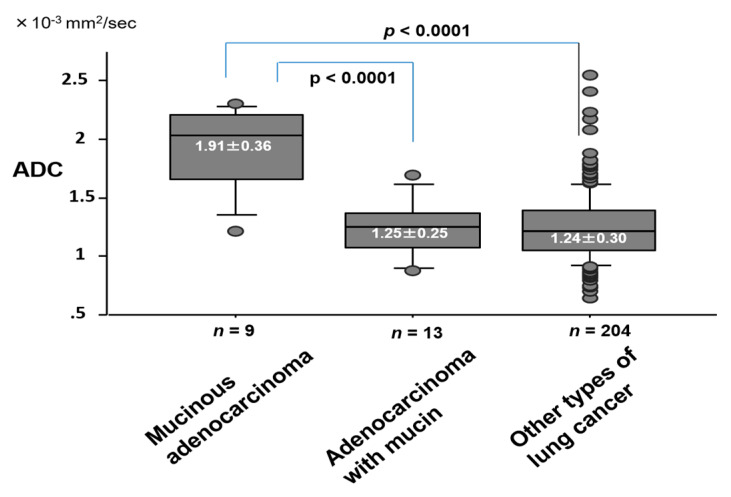
ADC value by the presence of mucin component in lung cancer.

**Figure 8 cancers-12-01194-f008:**
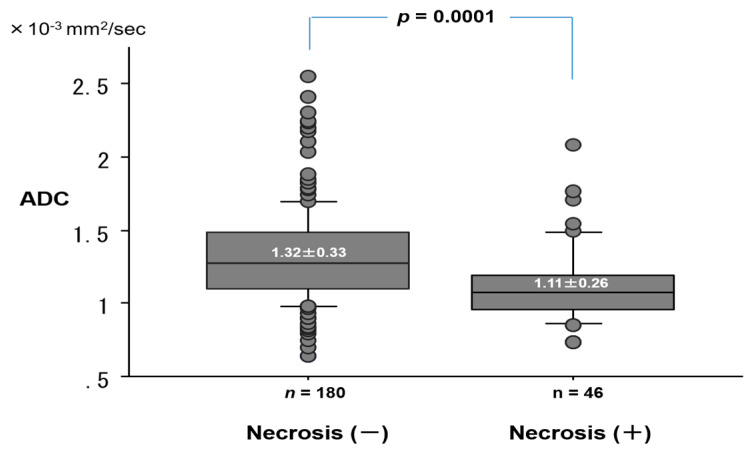
ADC value by the presence of necrosis in lung cancer.

**Table 1 cancers-12-01194-t001:** Sensitivity of diffusion-weighted imaging (DWI) for lung cancer.

Cell Type	Cell Subtype	Sensitivity	
Adenoca.	Mucinous	33.3% (3/9)	91.6% (153/167)
	Acinar	94.4% (51/54 )	
	Papillary	94.6% (53/56)	
	Lepidic	92.8% (26/28)	
	Micropapillary	100% (7/7)	
	Solid	100% (12/12)	
Squamous cell ca.			95.4 % (42/44)
LCNEC			66.6% (2/3)
Large cell ca.			66.6% (2/3)
Small cell ca.			100% (6/6)
Other cell type			100% (4/4)
Total			92.0% (208/226)

**Table 2 cancers-12-01194-t002:** Patients’ characteristics.

Age	37–85	Mean 68
Sex	Male	133
	Female	93
Cell Type	Adenoca.	167
	Squamous cell ca.	44
	Small cell ca.	5
	LCNEC	3
	Large cell ca.	3
	Other cell types	4
Operation	Pneumonectomy	7
	Bilobectomy	3
	Lobectomy	178
	Segmentectomy	2
	Partial resection	36
pN	N0	179
	N1	30
	N2	17
Pathological Stage	IA	112
	IB	49
	ⅡA	20
	ⅡB	13
	ⅢA	25
	ⅢB	1
	Ⅳ	6
Cell Differentiation	Well differentiated	109
	Moderately differentiated	73
	Poorly differentiated	37
	Undifferentiated	7

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
