# Peer review of "Relationships and Qualitative Evaluation between Diffusion-Weighted Imaging and Pathologic Findings of Resected Lung Cancers"

_cancers, 2020, doi:10.3390/cancers12051194_

Round 1

Reviewer 1 Report

Thank you for the opportunity to review this interesting study.

The authors have investigated a large patient sample of 226 patients with lung cancer undergoing DWI for the discrimination purposes.

A cut off value of 1.7 mm²/s was used, which was proposed by the same group in 2011. DWI achieved a sensitivity of 92%.

It is an interesting study with a timely topic.

Yet, I have some concerns to address:

The patient cohort was recruited from 2009 to 2014 with an older 1.5 T scanner. I have some concernings that patients were already reported in other studies, the authors cite in the references. Please specifiy this in the material and methods part, if you already used the patients elsewhere.

You did only provide a single slide measurement. Could you perform a whole lesion measurement, which might better characterize the tumors?

Why did the radiologist and pulmonologist performed the measurement and a consensus was reached? Could you perform an interreader variability for a sub-set of patients to show the reliability of DWI?

Did you exclude some patients due to lower imaging quality?

There are some imaging artifacts to see on the ADC of figure 2 and DWI of figure 4. This might be a limitation of this technique compared to PET.

Please provide box plots instead of bar graphs for better visualization.

The principle finding of the present study is that mucinous carcinoma have higher ADC values than the proposed cut off value. This is a already known fact for breast DWI with very similar appearance.

Interestingly, cancers with necrosis have lower ADC values than the ones without. This finding is contrary to first believe, as it is stated in many papers that necrosis leads to elevated ADC values. Please discuss this finding more.

In my opinion, it is more important in clinical routine to savely address benign finding as to detect malignancy in lung cancer. Benign findings with elevated FDG-PET uptake are lesions, which must be better characterized in clinical routine because these are unnecessarily biopted. The present study just investigated malignant lesions and no benign findings which suffers from a selection bias.

Minor concerns:

materials and methods: 4.4. pathological finding and statistical analysis

Disccusion: please correct FDG in the first sentence.

Author Response

Author's Reply to the Review Report (Reviewer 1)

Comments and Suggestions for Authors

Thank you for the opportunity to review this interesting study.

The authors have investigated a large patient sample of 226 patients with lung cancer undergoing DWI for the discrimination purposes.

A cut off value of 1.7 mm²/s was used, which was proposed by the same group in 2011. DWI achieved a sensitivity of 92%.

Point 1: It is an interesting study with a timely topic. Yet, I have some concerns to address:

The patient cohort was recruited from 2009 to 2014 with an older 1.5 T scanner. I have some concernings that patients were already reported in other studies, the authors cite in the references. Please specifiy this in the material and methods part, if you already used the patients elsewhere.

Response 1:

I reported an assessment of pulmonary nodules and masses between May 2009 and February 2012 [26. Usuda K, et al. Diagnostic performance of diffusion weighted imaging of malignant and benign pulmonary nodules and masses: comparison with positron emission tomography. Asian Pac J Cancer Prev. 2014;15:4629–35]. The previous article [26] dealt with 189 patients with pulmonary nodules and masses prospectively that underwent FDG-PET and DWI. There were 143 lung cancers, 17 metastatic lung tumors, and 29 benign pulmonary nodules and masses. The 143 lung cancers were included in this study.

Point 2: You did only provide a single slide measurement. Could you perform a whole lesion measurement, which might better characterize the tumors?

Why did the radiologist and pulmonologist performed the measurement and a consensus was reached? Could you perform an interreader variability for a sub-set of patients to show the reliability of DWI?

Response 2:

Several slices of DWI and ADCmap were prepared based on the size of lung cancer. This project were carried out mainly of the radiologist and the pulmonologist. The procedure of ADC measurement was repeated three times and the minimum ADC value was obtained. For everyday clinical examinations, an assessment of ADC measurement in the maximal sectioned surface is enough although a whole lesion measurement would be ideal it was impossible due to limitations in time. Regretfully an interreader variability was not assessed in this study although they eventually reached at the same consensus.

Point 3: Did you exclude some patients due to lower imaging quality?

There are some imaging artifacts to see on the ADC of figure 2 and DWI of figure 4. This might be a limitation of this technique compared to PET.

Response 3: There were 3 patients who were excluded due to lower imaging quality. This might be a limitation of this technique compared to PET. I added it to the limitations of DWI.

Point 4: Please provide box plots instead of bar graphs for better visualization. 

Response 4: We provided box plots in Figure 5-8 instead of bar graphs for better visualization.

Point 5: The principle finding of the present study is that mucinous carcinoma have higher ADC values than the proposed cut off value. This is a already known fact for breast DWI with very similar appearance.

Interestingly, cancers with necrosis have lower ADC values than the ones without. This finding is contrary to first believe, as it is stated in many papers that necrosis leads to elevated ADC values. Please discuss this finding more.

Response 5: In fact, pulmonary mucinous carcinomas were usually hypointense and showed higher ADC values, which could be misdiagnosed as benign lesions in DWI [26]. Mucinous carcinomas had lower DWI signal intensity and higher ADC values than tubular adenocarcinoma in the ano-rectal region, because mucinous carcinomas present lower cellularity than tubular adenocarcinomas [34].

One of the pulmonary lesions which presented restricted diffusion and lower ADC values in DWI was a pulmonary abscess with pathological necrosis. The heavily impeded mobility of pus may be caused by its high cellularity and viscosity, and shows the low ADC values [30]. Abscesses and thrombi impede the diffusivity of water molecules because they have a hyperviscous nature [31,32]. Median ADC values (0.877 x 10-3mm2/sec) of abscess were significantly lower than that (2.118 x 10-3mm2/sec) of phlegmon(p<0.001), and than that (3.008 x 10-3mm2/sec) of edema(p<0.01) [38]. In our article, ADCs of lung cancers were distributed over 0.9 to 1.6 x 10-3mm2/sec. At the result, ADC of a lung cancer with necrosis became lower than that of a lung cancer without necrosis because the ADC value of abscess was lower. If ADC value of another tumor was lower than that of the abscess, ADC of a tumor with necrosis would become higher than that of a tumor without necrosis.  

In DWI, 22% of benign lesions were revealed to be able to exhibit restricted diffusion in images with high b values [33]. The articles for the characteristics of abscesses and thrombi can show false-positive results in DWI for some benign pulmonary nodules and masses with abscesses.

Point 6: In my opinion, it is more important in clinical routine to savely address benign finding as to detect malignancy in lung cancer. Benign findings with elevated FDG-PET uptake are lesions, which must be better characterized in clinical routine because these are unnecessarily biopted. The present study just investigated malignant lesions and no benign findings which suffers from a selection bias.

Response 6: I agree with the reviewer’s opinions. We should add DWI for the differential diagnosis of benign lesions and malignant lesions of not only lung but also other organs. I think the differential diagnosis would be possible when we understand strengths and weaknesses of DWI.

Point 7: Minor concerns:

materials and methods: 4.4. pathological finding and statistical analysisDisccusion: please correct FDG in the first sentence.

Response 7:  We corrected them as 18-fluoro-2-deoxy-glucose positron emission tomography / computed tomography (FDG-PET/CT)

Reviewer 2 Report

The manuscript entitled "Relationships and Qualitative Evaluation between Diffusion-Weighted Imaging and Pathologic Findings of Resected Lung Cancers" focused the attention on the role of DWI for the evaluation of lung cancer.

  • The Introduction section is too short and should be expanded.
  • Could the Authors better explain how they recognize the optimal cutoff value?
  • The Authors reported the role of DWI in malignant nodules. Should they reported their experience, in term of sensitivity abnd specificity, also in benign nodules?

Author Response

Author's Reply to the Review Report (Reviewer 2)

Comments and Suggestions for Authors

The manuscript entitled "Relationships and Qualitative Evaluation between Diffusion-Weighted Imaging and Pathologic Findings of Resected Lung Cancers" focused the attention on the role of DWI for the evaluation of lung cancer.

Point 1: The Introduction section is too short and should be expanded.

Response 1:

In the introduction, I added several sentences for better understanding.

Lung cancer is one of the leading causes of cancer-related deaths and has many patterns of progression and treatment responses. Positron emission tomography with 18-fluoro-2-deoxy-glucose (FDG-PET) has been widely adopted as the imaging method of choice in tumor staging. The maximum standardized uptake value (SUVmax) is a parameter of glucose uptake and usually indicates how aggressive the cancer is. FDG-PET/CT has helped differentiate malignant from benign pulmonary nodules [1]. However, FDG-PET/CT is likely to give false-negative results for small volumes of metabolically active tumors [2], or well-differentiated pulmonary adenocarcinoma [3], and false-positive results for inflammatory nodules [4].

For the last two decades, magnetic resonance imaging (MRI) of staging of lung cancer has been limitedly available in the cases of chest wall invasion or mediastinum invasion of lung cancer partly due to the report of Webb et al [5] of the Radiologic Diagnostic Oncology Group in 1991. The Technology of MRI has developed dramatically. Diffusion-weighted magnetic resonance imaging (DWI) has been applied for detecting the restricted diffusion of water molecules. The principals of DWI takes the random motion of water molecules in biological tissues [6]. Its apparent diffusion coefficient (ADC) value presents a quantitative parameter of the diffusion of water molecules in biological tissues, and the ADC of malignant tumors is significantly lower than that of normal tissues or benign lesions [7].The MR signal intensity of pulmonary cancer nodules was significantly higher than that of benign lesions [8]. A meta-analysis has shown that DWI can be used to differentiate malignant from benign pulmonary lesions [9]. Two articles of meta-analysis reported that DWI was effective for the evaluation of N factor of lung cancer [10,11]. Peerlings et al. [10] reported high diagnostic capability of DWI for nodal assessment in non-small cell lung cancer: The sensitivity was 0.87 and the specificity 0.88. DWI can distinguish benign from malignant lesions in the lung [9, 12], in the thorax [13], in the prostate [14], in the breast [15], and in the liver [16].

DWI possess great potential for monitoring treatment response in cancer patients shortly after the initiation of radiotherapy [17]. Functional evaluation of DWI was more useful than that of CT for the response evaluation of chemotherapy and/or radiotherapy to recurrent tumors of the lung [18]. MR functional imaging offers valuable information about tumor tissue, tissue architecture, cellular biomarkers related to the hepatocellular functions, and tissue vascularization profiles related to tumor and tissue biology [19]. Maximum whole tumor ADC values may be available for differentiating luminal from other molecular subtypes of breast cancer [20].

In this article, we analyzed relationships between DWI and pathological findings of resected lung cancers and got information about DWI for pathologic characteristics of lung cancer.

In the discussion, I added our reported data for the quality of DWI.

There were two articles which compared diagnostic capability of DWI with that of FDG-PET/CT for pulmonary nodules and masses [12, 26]: The sensitivity and the accuracy of DWI were significantly higher [12], or the sensitivity of DWI was significantly higher [26] than those of FDG-PET/CT. DWI was reported to be superior to FDG-PET in the detection of primary lesions and the nodal assessment of non-small cell lung cancers [21].

Point 2: Could the Authors better explain how they recognize the optimal cutoff value?

The Authors reported the role of DWI in malignant nodules. Should they reported their experience, in term of sensitivity and specificity, also in benign nodules?

Response 2:

The optimal cutoff value is very useful for distinguishing malignancy from benignity but would be changed based on patients of a study. We reported an assessment of DWI for pulmonary nodules and masses [26]. 143 lung cancers, 17 metastatic lung tumors, and 29 benign pulmonary nodules and masses were assessed in this study. The sensitivity (80.0%) of DWI was significantly higher than that (70.0%) of FDG-PET. The specificity (65.5%) of DWI was equal to that (65.5%) of FDG-PET. The accuracy (77.8%) of DWI was not significantly higher than that (69.3%) of FDG-PET for pulmonary nodules and masses. DWI has higher potential than PET in assessing pulmonary nodules and masses.

In our experience of hilar and mediastinal lymph node in lung cancer, [Usuda K, et al. Ann Surg Oncol 20; 1676-1683: 2013.] DWI correctly diagnosed N staging in 144 carcinomas (90 %) but incorrectly diagnosed N staging in 16 (10 %) [3 (1.9 %) had overstaging, 13 (8.1 %) had understaging]. PET-CT correctly diagnosed N staging in 133 carcinomas (83.1 %) but incorrectly diagnosed N staging in 27 (16.8 %) [4 (2.5 %) had overstaging, 23 (14.4 %) had understaging]. The maximum diameter of metastatic lesions in lymph nodes were 3.0 ± 0.9 mm in 21 lymph node stations not detected by either DWI or PET-CT: 7.2 ± 4.1 mm in 39 detected by DWI, and 11.9 ± 4.1 mm in 24 detected by PET-CT. DWI could detect significantly smaller lymph node metastases than PET-CT could.

Round 2

Reviewer 1 Report

Thank you for the revision of the manuscript. I think it is suitable for publication in the current form.

Reviewer 2 Report

I have no further comments. 

This manuscript is a resubmission of an earlier submission. The following is a list of the peer review reports and author responses from that submission.